# Analysis of the Common Femoral Artery and Vein: Anatomical Morphology, Vessel Relationship, and Factors Affecting Vessel Size

**DOI:** 10.3390/medicina58020325

**Published:** 2022-02-21

**Authors:** Sang-hun Lee, Dong uk Yu, Tae kwon Kim, Jae-cheon Jeon, Sang chan Jin, Woo ik Choi, Jae ho Lee

**Affiliations:** 1Department of Emergency Medicine, Dongsan Medical Center, Keimyung University School of Medicine, 1035, Dalgubeol-daero Dalseo-gu, Daegu 42601, Korea; donguk932@naver.com (D.u.Y.); jason_williams@daum.net (T.k.K.); cheon9803@naver.com (J.-c.J.); jchan98@daum.net (S.c.J.); emtaegu@dsmc.or.kr (W.i.C.); 2Department of Anatomy, Keimyung University School of Medicine, 1035, Dalgubeol-daero Dalseo-gu, Daegu 42601, Korea; anato82@dsmc.or.kr

**Keywords:** femoral artery, femoral vein, computed tomography

## Abstract

*Background and Objectives*: We aimed to analyze the morphology of the common femoral artery (CFA) and common femoral vein (CFV) and the anatomical relationship between the two blood vessels, and to investigate the factors that influence the size of these blood vessels. *Materials and Methods*: This retrospective study included 584 patients who underwent abdominal and pelvic computed tomography from 1 February to 28 February 2021. We measured the vessels at three regions on both lower extremities (inguinal ligament, distal vessel bifurcation, midpoint) and analyzed and classified the degree of overlap between the CFA and CFV into three types, as well as the factors affecting vessel size. *Results*: After comparing the femoral vessels according to location, it was confirmed that the CFA and CFV were larger distally than proximally on both sides (*p* < 0.001). The degree of overlap increased distally (*p* < 0.001) but was less at the middle (*p* < 0.001) and distal (*p* = 0.011) regions on the right side. It was found that the size of CFA and CFV were related to age, sex, and body mass index (BMI) and that malignancy also affects the CFA size. *Conclusions*: The morphology of the CFA and CFV was conical and increased distally. The degree of overlap between the two blood vessels also increased distally but was less on the right than on the left. Age, sex, and BMI are significant factors affecting the sizes of the CFA and CFV, and malignancy is associated with the CFA size.

## 1. Introduction

Aside from transporting blood, the common femoral artery (CFA) and common femoral vein (CFV) act as conduits for resuscitative fluids, blood products, drugs, total parenteral nutrition administration, vascular intervention procedures, such as endovascular procedures or cardiovascular interventions, and mechanical extracorporeal therapies, such as renal replacement therapy and extracorporeal membrane oxygenation [1,2]. The vessel of choice of the various procedures will depend on the patient’s condition and the judgment of physicians. To perform invasive procedures, both the CFA and CFV have few mechanical side effects and are easily accessible, so they are often used in critical care patients [3,4,5]. Emergency vascular access is time-sensitive, and early successful vascularization significantly impacts patient prognosis [2]. Therefore, clinicians are encouraged to be aware of the patient’s ideal anatomical vessel location, which can be useful in invasive procedures using CFA and CFV.

The CFA is located on the lateral side of the CFV, and the degree of overlap between the two blood vessels increases as they course distally from the inguinal ligament [6]. Various studies have measured the size of the femoral vessels, and factors, such as age, sex, height, weight, body surface area, and underlying diseases, have been identified that affect their size [7,8]. As most blood vessels are oval rather than circular, their circumference must be obtained to measure their size accurately. However, previous studies only measured the diameter of blood vessels. The regions where the CFA and CFV were measured were not consistent, and morphological analysis was not performed on anatomically subdivided vessel regions. In addition, the results of previous studies for factors affecting blood vessel size were different, and the number of participants enrolled in each study was relatively small.

In this study, we investigated the anatomical morphology of the CFA and CFV according to subdivided regions in patients who underwent abdominal and pelvic computed tomography (APCT) and evaluated the factors affecting their size.

## 2. Materials and Methods

### 2.1. Study Design and Search Strategy

This retrospective study included adult patients aged ≥18 years who underwent contrast-enhanced APCT at the Keimyung University Dongsan Hospital of Daegu, Korea from 1 February to 28 February 2021. Patients without the required data for the study, such as height, weight, and records of their underlying disease, or those who had poor CT images were excluded. Patients who had previously undergone surgery on the pelvis or femoral vessels, or those who had peripheral vascular disease or severe blood vessel calcification were also excluded. To properly divide the CFA and CFV into proximal, middle, and distal regions, patients with a short vessel length (≤2 cm) were excluded.

### 2.2. Data Collection

Patient characteristics, including age, sex, height, and weight were collected from electronic medical records. Additionally, history of previous illnesses, including hypertension, diabetes, dyslipidemia, chronic kidney disease, cerebrovascular accident, coronary artery disease, pulmonary thromboembolism, malignancy, history of chemotherapy, and history of radiotherapy, was collected.

The blood vessels taken by CT were measured along the circumference of the blood vessel wall, and the size of the area inside the blood vessels was evaluated. The circumferences of the CFA and CFV were measured at three regions: the inferior margin of the inguinal ligament (proximal), the origin of vessel bifurcation to the deep femoral vessel (distal), and the midpoint between the proximal and distal regions (middle). The degree of overlap between the CFA and CFV was classified into three types: Type I, in which the CFA and CFV do not overlap; Type II, in which the CFA overlaps the CFV, but the lateral part of the CFV does not exceed half of the width of the CFA; and Type III, in which the CFA overlaps the CFV, and the lateral part of the CFV exceeds half of the CFA. 

### 2.3. Statistical Analysis

Non-normally distributed continuous variables, as determined using the Kolmogorov–Smirnov test, were reported as medians and interquartile ranges and compared using the Mann–Whitney *U* test. To compare vessel regions (proximal, middle, and distal), we used an analysis of variance with post hoc corrections using the Kruskal–Wallis method. Categorical variables were reported as numbers and percentages and compared using the χ^2^ or Fisher’s exact test, as appropriate. Continuous variables were changed to categorical variables to perform a statistical regression analysis of the factors affecting vessel size. According to the median size, the vessel sizes were divided into two groups (large and small). After dividing into groups according to the measured sizes at the distal levels of the CFA and CFV, univariate logistic regression analysis was performed to determine any association with the demographics and clinical characteristics. Variables were adjusted for age, sex, hypertension, diabetes, dyslipidemia, chronic kidney disease, coronary artery disease, cerebrovascular events, malignancy, history of chemotherapy, history of radiotherapy, and pulmonary thromboembolism. Large and small CFA and CFV sizes were analyzed using multivariate logistic regression analysis, and the results are reported as odds ratios (ORs) and 95% confidence intervals (CIs). Variables were tested for goodness-of-fit using the Hosmer–Lemeshow method. Two-sided *p* values < 0.05 were considered statistically significant. All statistical analyses were performed using SPSS version 25.0 for Windows (SPSS Inc., Armonk, NY, USA).

### 2.4. Ethics Statement

The study protocol was approved by the Institutional Review Board of Keimyung University Dongsan Hospital (no. 2021-10-008), which waived the requirement for informed consent owing to the retrospective nature of this study.

## 3. Results

During the study period, 1190 patients underwent APCT; 606 patients were excluded from the study due to predetermined criteria, including 25 (4.1%) younger than 18 years, 271 with missing data (44.7%), 41 with previous operations in the abdominal or pelvic area (6.8%), 23 with poor diagnostic images (3.8%), 3 with peripheral vessel disease (0.5%), 35 with severe vessel calcification (5.8%), and 208 (34.3%) with short CFA or CFA that could not be divided into grades (Figure 1).

A total of 584 patients were enrolled: 210 with hypertension (36.0%), 151 with diabetes (25.9%), 54 with dyslipidemia (9.2%), 22 with chronic kidney disease (3.8%), 34 with cerebrovascular accidents (5.8%), 60 with coronary artery disease (10.3%), 12 with pulmonary thromboembolism (2.1%), 367 with malignancy (62.8%), 231 with a history of chemotherapy (39.6%), and 64 with a history of radiotherapy (11.0%). Among them, 340 were men (58.2%), and the medians of age, height, weight, and body mass index (BMI) of all patients were 65 years, 163.0 cm, 60.7 kg, and 23.3 kg/m^2^, respectively. By gender, it was found that the men were older than the women, and their heights and weights were higher. Additionally, diabetes, coronary artery disease, and malignancy were more common in men than in women (Table 1).

After comparing the CFA and CFV, it was confirmed that the distal regions were larger than the proximal regions on both sides. The median sizes of the CFA and CFV at the proximal, middle, and distal regions were 61.9 mm^2^, 66.9 mm^2^, and 73.1 mm^2^, and 97.5 mm^2^, 111.0 mm^2^, and 126.6 mm^2^, respectively (both *p* < 0.001). No size differences were observed between the CFA and CFV on both sides. The degree of overlap between the CFA and CFV was observed to increase distally. Types I, II, and III were observed in 90.8%, 8.7%, and 0.5% of the proximal regions, 79.1%, 19.2%, and 1.7% in the middle regions, and 58.2%, 35.9%, and 5.9% in the distal regions, respectively (*p* < 0.001). On the right side, more type I but fewer type II and III were observed than on the left side in the middle (*p* < 0.001) and distal (*p* = 0.011) regions (Table 2) (Figure 2).

In this study, the distal CFA and CFV measured the largest, and the right side had less overlap; therefore, regression analysis was performed on the right distal CFA and CFV. Table 3 shows the results of the multivariate analysis for the factors associated with the CFA and CFV areas divided into two groups according to their median sizes (CFA: 73.1 mm^2^; CFV: 126.6 mm^2^). Multivariate regression analysis was performed using the adjusted variables, including age, sex, hypertension, diabetes, dyslipidemia, chronic kidney disease, coronary artery disease, cerebrovascular event, malignancy, history of chemotherapy, history of radiotherapy, and pulmonary thromboembolism. According to the results, advanced age (OR: 1.045; 95% CI: 1.030–1.060), male sex (OR: 4.956; 95% CI: 3.340–7.354), higher BMI (OR: 1.176; 95% CI: 1.116–1.240), and malignancy (OR: 2.136; 95% CI: 1.414–3.227) were factors significantly associated with right distal CFA size. Factors related to right distal CFV size were advanced age (OR: 1.032; 95% CI: 1.018–1.047), male sex (OR: 5.385; 95% CI: 3.625–7.999), and higher BMI (OR: 1.231; 95% CI: 1.166–1.301).

## 4. Discussion 

This study found that the CFA and CFV did not significantly differ between the right and left lower extremities and that these vessels had a conical morphology that increased in size as it coursed distally. The degree of overlap of the CFA and CFV was minimal in the proximal region but increased towards the distal region; however, there was less overlap on the right side. The CFA and CFV were larger in men, those with advanced age, and those with a high BMI; the CFA was also larger when malignancy was present.

Anatomically, the CFA is the continuation of the external iliac artery. It begins at the inguinal ligament and ends at the bifurcation of the deep femoral artery. It is generally known that the CFV is located more medially than the CFA [9]. Various methods to assess the size of the CFA and CFV have been performed. However, these measured the vessel diameter, and the measured vessel location was anatomically unclear [7,8,10]. Since blood vessels are oval rather than circular, assessing a vessel’s size by measuring its diameter has limited accuracy in assessing the size. Moreover, blood vessels do not maintain the same size throughout their course, and their size varies depending on the anatomical position. 

Hence, a comparative evaluation based on the size of blood vessels measured at unspecified positions is unacceptable. To correct the limitations of previous studies, this study divided the CFA and CFV into three regions (proximal, medial, and distal), and the size of the vessel at each region was measured to more accurately assess vessel size. To the best of our knowledge, this is the first study to present the shapes of the CFA and CFV and compare their sizes based on their areas rather than their diameters. The results show that the shapes of both the CFA and CFV were conical and that their sizes were not constant but rather increased distally. The shapes of both the CFA and CFV were similar on both sides. There was also no difference in the size and anatomical morphology between both sides, similar to the results of previous studies [8,11].

It is known that the degree of overlap between the CFA and CFV increases distally [2]. A previous study showed that the common femoral vessels on the right side have relatively less overlap than those on the left side [12]. However, few studies have analyzed the morphology of blood vessels by measuring the sizes and degrees of overlap between the CFA and CFV. In this study, the degree of overlap between the CFA and CFV varied according to the location on the vessel. A 9.2% overlap between the two vessels was observed in the proximal region, and no difference was observed between both sides. However, a higher degree of overlap was observed not only on the middle and the distal regions but also on the left more than on the right side.

Vascular catheterization through the CFA and CFV should be performed carefully. If the needle is introduced at a location more superior than the proximal portion of the CFA or CFV, it may puncture the external iliac vessels, which are located at the superior inguinal ligament, or it may penetrate them and enter the retroperitoneal space [13]. Retroperitoneal space injuries, such as hematomas, are extremely fatal and difficult to manage because of the difficulty in achieving compression of the bleeding vessel [14]. Conversely, blood vessels, such as the superficial or deep femoral vessels, may be positioned at a location more inferior than the distal portion of the CFA or CFV [15,16]. If a needle is introduced at this location, then not only do these vessels reduce the probability of successful catheter placement but also increase the risk of complications such as occlusion, thrombosis, and distal leg ischemia [17]. The CFA’s length is about 4–5 cm; however, it has been observed to be as short as 2 cm. Therefore, it is challenging to secure a blood vessel at an ideal position using traditional methods [7]. Recently, ultrasonography has become easier to perform, which has made it possible to accurately confirm the anatomical location of blood vessels, thereby improving the probability of catheterization and decreasing the risk of complications [5,18]. As discussed, the sizes and degree of overlap of the CFA and CFV increase distally, and the overlap on the right side is less than on the left. Therefore, when using ultrasonography to visualize the CFA and CFV, we recommend first approaching the distal region on the right side.

The CFA and CFV size-related factors include age, sex, and BMI; other central vessels have also been found to have the same factors affecting vessel size [8,11,19,20]. The morphology of blood vessels changes with age, and their components, made up of proteins, elastin, and collagen, also deform. As a person ages, blood vessels lose their elasticity due to long-term repetitive stretching, and collagen becomes stiffer, which is thought to increase the size of blood vessels [21]. Athletes with high muscle mass need more blood supply to the skeletal muscles than ordinary people. As the body responds to the increased demand, blood vessels become larger [22]. Since men generally have more muscle than women, more blood supply is needed, and the CFA and CFV also increase in size accordingly [23]. Obesity increases intra-abdominal pressure; higher vascular pressure is needed for the required blood circulation, which increases the sizes of the femoral vessels [24]. This study shows that age, sex, and BMI affect femoral vessel size, whereas malignancy affects the size of the CFA. In malignancy, angiogenesis, vessel remodeling, and increased permeability increase blood supply to the tumor, increasing the size of the CFA [25].

This study has several limitations. First, this study had a retrospective single-center design. Considering the retrospective nature, not all patients could provide accurate information, such as previous illness history, medication history, and physical status, which may not have been fully described in the medical records, leading to data abstraction errors. Second, we did not include an analysis of the use of volume status, vital signs, and vasopressors, which are factors that may affect the size of a patient’s blood vessels at the time of the CT scan. Third, it was impossible to determine the overall vessel state by measuring only partial vessel sizes without measuring all successive position levels of the common femoral vessels. Fourth, common femoral vessels shorter than 2 cm and various anatomical vessel variations were not analyzed.

## 5. Conclusions

The CFA and CFV are conical, and their sizes increased from the proximal to the distal region. Distally, there was a greater degree of overlap between the two vessels, and the overlap on the right side was less than that on the left side. Age, BMI, and sex influenced the sizes of the CFA and CFV, and malignancy influenced the size of the CFA. It is expected that the results of this study will enable the selection of the ideal positions for the approach of the CFA and CFV, increasing the success rate of the procedure and reducing the occurrence of complications.

## Figures and Tables

**Figure 1 medicina-58-00325-f001:**
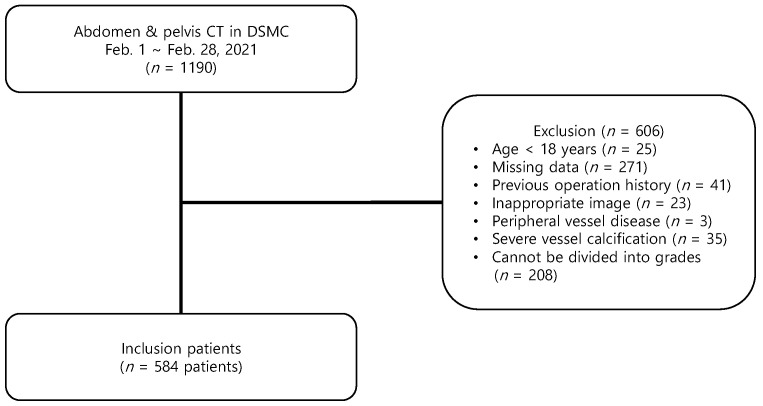
Flowchart of the study patients.

**Figure 2 medicina-58-00325-f002:**
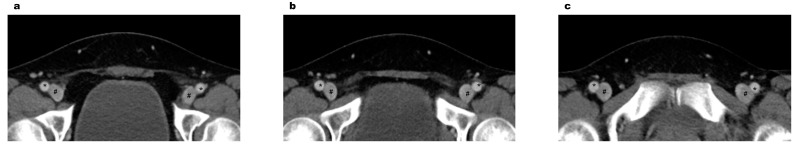
Abdominal and pelvic computed tomography axial images of the common femoral artery (*), common femoral vein (^#^), and surrounding structures in the pelvis. (**a**) Proximal location, (**b**) middle location, (**c**) distal location.

**Table 1 medicina-58-00325-t001:** Demographic and clinical characteristics.

Patient Characteristics	Total (*n* = 584)	Men (*n* = 340)	Women (*n* = 244)	*p* Value
Age, years	65 (56–73)	66 (59–73)	64 (50–72)	0.016
Height (cm)	163.0 (156.9–169.0)	168.0 (163.0–172.0)	156.0 (150.6–160.1)	<0.001
Weight (kg)	60.7 (53.9–69.6)	65.0 (56.9–73.1)	55.5 (50.9–62.8)	<0.001
Body mass index (kg/m^2^)	23.3 (20.9–25.7)	23.3 (20.9–25.6)	23.1 (20.9–25.9)	0.859
Previous illness				
Hypertension	210 (36.0)	125 (36.8)	85 (34.8)	0.348
Diabetes	151 (25.9)	101 (29.7)	50 (20.5)	0.008
Dyslipidemia	54 (9.2)	31 (9.1)	23 (9.4)	0.505
Chronic kidney disease	22 (3.8)	12 (3.5)	10 (4.1)	0.442
Cerebrovascular accident	34 (5.8)	22 (6.5)	12 (4.9)	0.273
Coronary artery disease	60 (10.3)	46 (13.5)	14 (5.7)	0.001
Pulmonary thromboembolism	12 (2.1)	4 (1.2)	8 (3.3)	0.072
Malignancy	367 (62.8)	225 (66.2)	142 (58.2)	0.030
History of Chemotherapy	231 (39.6)	132 (38.8)	99 (40.6)	0.366
History of Radiotherapy	64 (11.0)	43 (12.6)	21 (8.6)	0.079

**Table 2 medicina-58-00325-t002:** Comparison of femoral artery and vein sizes by region.

Location	Total (*n* = 1168)	Right (*n* = 584)	Left (*n* = 584)	*p* Value
Femoral artery size (mm^2^)				
Proximal	61.9 (49.8–74.8)	62.3 (50.4–75.0)	61.6 (49.6–74.1)	0.306
Middle	66.9 (55.0–81.4)	67.0 (55.1–82.7)	66.9 (54.8–81.1)	0.592
Distal	73.1 (58.7–88.7)	73.7 (59.8–90.4)	72.0 (58.2–87.5)	0.151
*p* value	<0.001	<0.001	<0.001	
Femoral vein size (mm^2^)				
Proximal	97.5 (74.5–122.2)	97.1 (75.1–122.9)	97.9 (74.3–122.1)	1.000
Middle	111.0 (81.6–147.2)	109.9 (81.8–147.6)	113.5 (80.8–147.9)	0.944
Distal	126.6 (91.1–169.2)	123.8 (90.4–164.2)	128.8 (91.8–173.7)	0.219
*p* value	<0.001	<0.001	<0.001	
Morphology types between veins and arteries				
Proximal				0.206
Type I	1060 (90.8)	534 (91.4)	526 (90.1)	
Type II	102 (8.7)	48 (8.2)	54 (9.2)	
Type III	6 (0.5)	2 (0.3)	4 (0.7)	
Middle				<0.001
Type I	924 (79.1)	491 (84.1)	433 (74.1)	
Type II	224 (19.2)	84 (14.4)	140 (24.0)	
Type III	20 (1.7)	9 (1.5)	11 (1.9)	
Distal				0.011
Type I	680 (58.2)	359 (61.5)	321 (55.0)	
Type II	419 (35.9)	195 (33.4)	224 (38.4)	
Type III	69 (5.9)	30 (5.1)	39 (6.7)	
*p* value	<0.001	<0.001	<0.001	

**Table 3 medicina-58-00325-t003:** Multivariate regression analysis of factors affecting the common femoral artery and vein.

Variables	Crude OR	95% CI	Adjusted OR	95% CI
Common femoral artery				
Age	1.038	1.026–1.051	1.045	1.030–1.060
Men	4.324	3.036–6.157	4.956	3.340–7.354
Body mass index	1.105	1.057–1.155	1.176	1.116–1.240
Malignancy	2.246	1.592–3.168	2.136	1.414–3.227
Common femoral vein				
Age	1.022	1.010–1.034	1.032	1.018–1.047
Men	4.380	3.075–6.239	5.385	3.625–7.999
Body mass index	1.171	1.115–1.229	1.231	1.166–1.301

CI, confidence interval.

## Data Availability

Not applicable.

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
