# Peer review of "Analysis of the Common Femoral Artery and Vein: Anatomical Morphology, Vessel Relationship, and Factors Affecting Vessel Size"

_medicina, 2022, doi:10.3390/medicina58020325_

Round 1

Reviewer 1 Report

The presented work serves to refinement of knowledge of data on the size of the proximal segment of the femoral artery and vein. Nevertheless, some improvements of the methods used are recommended:

In the Methods are not clearly visible (and in the text not defined) the proximal and distal borders of the measured vessels (i.e. accurate display of the inferior margin of inguinal ligament, and the origin of the deep femoral artery). The planimetric method used to measure the size of both vessels is not mentioned.

It is necessary to significantly refine the diagnoses of patients (lines 115-118), because it seems that in some cases there were combinations of them. Further, is needed to specify gender-based demographic data (height, weight, BMI etc.) in Table 1.

There are typing errors in the text that need to be corrected, for example “opertations” at line 109.

Anatomical remark: the authors do not mention, that used anatomical terms “common femoral artery“ and „common femoral vein“ are not official anatomical terms (see FIPAT. Terminologia Anatomica. Stuttgart: Thieme, 1998). The terms used are exact from the point of view of clinical anatomy and are also discussed in the literature (see Kachlik, D. et al. A plea for extension of the anatomical nomenclature: Vessels. Bosnian Journal of Basic Medical Sciences. 2021, 21(2), 208-220. DOI: 10.17305/bjbms.2020.5256.), but they are still not part of the current and prepared official anatomical terminology.

Reviewer 2 Report

The authors analyzed the morphometry of the CFA and CFV and identified factors that potentially affect their sizes. One of the strengths of this manuscript is the respectable number of patients. The rationale is original and clinical significance of the paper is relevant. 

Patients with short vessel length were excluded. What does it mean by this? Does it mean that these patients have some sort of variations, e.g. high bifurcation of the profunda femoris artery, or any other variants? If that’s the case, the authors may consider reporting these variants because they would be of importance in the field of anatomy.

The CFA and CFV were classified into three types. Is this based on any reference. If yes, please cite those references. 

Apart from the information in Table 3, the authors may consider plotting X-Y graphs to more clearly illustrate the effects of age and BMI on the two vessels (optional).

This is also optional but it would be more anatomically correct to use the term “femoral artery” and “femoral vein” without the term common, which are in line with Terminologia Anatomica. I will have to leave it to the authors to decide.
